# Clinical Strategies for Counteracting Human Ovarian Aging: Molecular Background, Update, and Outlook

**DOI:** 10.3390/ijms262411973

**Published:** 2025-12-12

**Authors:** Jan Tesarik, Raquel Mendoza Tesarik

**Affiliations:** MARGen (Molecular Assisted Reproduction and Genetics) Clinic, 18006 Granada, Spain

**Keywords:** ovarian aging, hypothalamic-pituitary-ovarian axis, antioxidant and mitochondrial treatment, hormonal and growth factor modulation, fertility preservation, precision medicine

## Abstract

Ovarian aging (OA) results from the senescence of different cell types present in the ovary, decreasing female fertility and quality of life and augmenting the risk of a variety of fertility-unrelated pathological conditions. The changes observed in the ovarian cells are accompanied by changes occurring in various elements of the hypothalamic–pituitary–ovarian (HPO) axis, the complex endocrine system that regulates the female reproductive cycle. Issues pertaining to the HPO axis have been addressed in animal models via hormonal treatments with preparations inhibiting ovarian follicular recruitment at the level of the receptors of gonadotropin-releasing hormone (GnRH)-secreting neurons, mainly acting on glutamate- and gamma-aminobutyric acid (GABA)-driven signaling. GnRH agonists and antagonists have also been used in women exposed to chemotherapeutics. HPO-independent OA can be delayed through the administration of different antioxidants and mitochondria-protecting agents, among which melatonin has been shown to be particularly useful. Other therapeutic approaches used with success in women include hormonal and growth factor (GF) modulators, such as growth hormone (GH), insulin-like growth factor 1 (IGF-1), vascular endothelial growth factors (VEGF), and dehydroepiandrosterone (DHEA), and the development of patient-tailored combination-based therapies (IGF-1 + VEGF + DHEA) has also been suggested. Intraovarian injection of autologous platelet-rich plasma (PRP), mitochondrial donation through pronuclear transfer, and ovarian tissue cryopreservation and autotransplantation have also yielded promising results in women, and their use can preserve not only fertility but also the ovarian endocrine function. Personalized mixtures of specific agents (desatinib, quercetin, rapamycin, metformin, resveratrol, melatonin, and coenzyme Q10) targeting different cell types in the ovary are currently under investigation. Overall, this review aims to present a global view of the subject, uniting the physiological and molecular background of this pathology with the history and development of potential treatment strategies and new perspectives in this domain. As such, this study may be helpful both to clinicians facing problems resulting from OA and to researchers pursuing further developments in this field.

## 1. Introduction

Human female reproductive aging is basically caused by ovarian aging (OA) because the uterine function, except for specific pathologies, remains unchanged even in postmenopausal women, as witnessed by the high, virtually age-independent live birth rates after the transfer of embryos resulting from oocytes donated by young women [1]. OA, marked by decreased fertility related to impaired oocyte quantity and quality, is a physiological process. However, in the context of the current trend of delaying motherhood because of different societal constraints, together with the fact that it may occur prematurely in a woman’s life [2], OA is the cause of serious emotional discomfort in women who wish to procreate. Beyond the fertility issues, several studies have reported the adverse effects of ovarian aging on different fertility-unrelated functions of the female body, entailing higher risks of almost every chronic age-related general health issue, such as osteoporosis, neurodegeneration, cardiovascular diseases [3], vasomotor symptoms (e.g., hot flashes and night sweats), and increased risk of diabetes and high blood pressure [4,5]. Thus, ovarian aging, along with the longer life expectancy of women than men, could be one of the underlying drivers of the so-called mortality–morbidity paradox, meaning that women usually live longer but less comfortably than men do [6]. This situation calls for urgent action to improve our current means of counteracting OA, both with respect to fertility preservation and to improving the general health and wellbeing of the affected women [7].

Many studies have addressed different aspects of OA in their search for more effective preventive and therapeutic methods. However, much is still to be done in both directions. Clearly, OA has a strong genetic and epigenetic component, as reviewed previously [2,8,9,10], which is specific for each woman, so any preventive and therapeutic actions taken need to be patient-tailored in the sense of modern precision medicine. This study briefly resumes the study of the basic characteristics of OA in women, which has recently been exhaustively reviewed [2,6,11,12,13], but with particular attention paid to findings that were not included in previous reviews, which may provide clues for the development of new therapies. Experimental animal data are mentioned only when useful for understanding the context of human clinical studies.

## 2. Basic Mechanisms of OA as Possible Therapeutic Targets

OA is characterized by two simultaneously occurring phenomena: the disruption of the hypothalamic–pituitary–ovarian (HPO) axis and the loss of ovarian follicle and oocyte quantity and quality (reviewed in Wang et al. [11]). Even though some earlier studies, performed in rodents, suggested that OA is primarily triggered by the disturbance of the HPO axis, with ovarian damage being its consequence [14,15], this view is currently not generally accepted; it is understood that HPO dysfunction may be a consequence rather than a cause of ovarian failure. However, finding the correct answer to this dilemma is of essential importance to determining the optimal strategies to use in OA therapy. In the absence of definite knowledge of the primary OA trigger, current therapies target both the HPO axis and the function of the ovarian cells (Table 1). These two strategies are further explained below (Section 2.1 and Section 2.2).

### 2.1. Targeting the HPO Axis

As in rodents, the HPO axis has also been shown to be altered in perimenopausal women. As compared to healthy young women, most aged women approaching menopause failed to generate an LH surge in response to estradiol challenge [16]. Moreover, a gradual increase in gonadotropin secretion has been observed to start as early as 27–28 years old and to accelerate dramatically after 49, concerning mainly FSH rather than LH [17]. Changes in gonadotropin secretion precede any detectable menstruation or ovulation failure and appear to result from an altered secretion pattern of the gonadotropin-releasing hormone GnRH [18,19].

Despite the absence of knowledge of the exact mechanism of the above neuroendocrine changes, circumstantial evidence suggests that they may somehow be related to abnormal neurotransmitter secretion in the central neural system and points to glutamate as being potentially responsible. In fact, the vesicular glutamate transporter 2 (VGLUT2), a protein that packages the excitatory neurotransmitter glutamate into synaptic vesicles to be released at synapses, has been localized by immunocytochemistry to the GnRH-secreting neurons in the preoptic region of the rat brain [20,21]. A decline in glutamate release from the preoptic area in aged female rats correlates with reduced GnRH secretion, postpones LH surge [15], and is accompanied by an attenuation of the expression of glutamate receptors in GnRH-secreting neurons [22].

In addition to glutamate, gamma-aminobutyric acid (GABA) and its GABAA receptor also play an important role in the regulation of GnRH neural secretory activity in rats [23], entering an intricate interplay with the glutamate receptors [24].

Both the glutamate- and GABA-driven signaling in GnRH neurons have been considered as potential targets of therapeutic action. In the rat model, researchers succeeded in restoring normal LH surge via combined treatment with a GABA antagonist and a glutamate agonist, suggesting that normal GnRH neural output can be reached by balancing the effects of excitatory and inhibitory neurotransmitters in the GnRH-secreting neurons [25].

Other neurotransmitters supposed to be involved in controlling the secretory activity of GnRH neurons include the peptides kisspeptin (K) [26], neurokinin B (N) [27], and dynorphin (Dy) [28]. These three peptides colocalize in the so-called KNDy cells—a group of neurons of the arcuate nucleus of the hypothalamus that co-express K, N, and Dy—and are conserved across a range of species from rodents to humans, and their interaction is believed to regulate both hypothalamic GnRH secretion and OA [28,29]. There is sufficient evidence indicating that the gene expression of these three peptides is altered in postmenopausal women and female monkeys, which is likely to cause an imbalance of excitatory and inhibitory input to GnRH neurons and thus to cause the neuroendocrine disorders involved in OA [11,30,31,32,33,34]. Whereas HPO axis (Figure 1) alterations may be the primary trigger of OA, it is also possible that it is a mere consequence of a decline in follicle numbers, leading to reduced ovarian hormone (mainly estradiol and inhibin) output and thus decreasing the negative feedback in pituitary FSH secretion (Figure 1), resulting in elevated serum FSH levels and exacerbating ovarian follicle depletion [11,34].

### 2.2. Targeting HPO-Independent Ovarian Decay

Compared to the relative scarcity of data on the HPO axis, HPO-independent ovarian decay has received much more attention across the latest decades, resulting in a variety of treatment strategies becoming clinically available. The depletion of ovarian follicles, resulting in hypoestrogenism, with all its negative health consequences, and the decrease in the quality of the remaining oocytes are the two principal consequences of OA [2,6,11,12]. However, in order to search for clinically effective prevention and treatment strategies, we have to ask how the different types of cells present in the ovary contribute to this process. The ovary is composed of multiple cell types, including oocytes, granulosa cells (GC), theca cells, stromal cells, immune cells, and smooth muscle cells. The relative contribution of each of these cell types to the overall picture of OA has been greatly facilitated since 2009, when single-cell RNA sequencing (scRNA-seq) was first used to analyze the transcriptome of mouse oocytes and embryos [35]. Since then, scRNA-seq has increasingly been used to analyze individual cell types in normal and aging ovaries (reviewed in Liang et al. [36]). By using this approach, a cascade of events—in which immune cell infiltration with stromal cell fibrosis worsens the ovarian microenvironment and GC apoptosis accelerates fertility loss, leading to reduced oocyte quality—has been mapped, along with the single-cell transcriptomic level [37,38,39].

It has long been known that oxidative stress, caused by the accumulation of reactive oxygen species (ROS), is intimately involved in the process of OA. ROS are by-products of normal cell (mitochondrial) metabolism, required for normal cell function, but their excessive accumulation within cells can have serious negative consequences, namely, follicle atresia and diminished oocyte quality [13]. This situation can result from an age-related decline in ROS scavenging, enhanced ROS production promoted by different lifestyle and environmental factors (see Section 4), or a combination of both. As to the former, it has been demonstrated that older women exhibit lower superoxide dismutase (SOD) levels in GC, causing decreased antioxidant efficacy with OA [13]. In addition to ROS, advanced glycosylation end-products (AGEs), formed by reactive carbonyl species and free amino groups gained from a series of nonenzymatic reactions [40,41], also contribute to OA [11].

As to the oxidative and mitochondrial stress caused by excessive ROS, an accumulation of the effects of negative lifestyle factors (see Section 3) is to be blamed. In order to compensate for this age-related condition, antioxidant and mitochondria-protective treatments, such as vitamins C, D, and E, melatonin, and others, have been used for decades to slow ovarian aging [42] (see Section 3 for the latest updates). Oocytes, as some of the most long-living cells in the organism, are particularly vulnerable to oxidative stress. The deficiencies of the mitochondrial function brought about by oxidative stress lead to damage to the proper mitochondrial DNA (mtDNA), further increasing the problem, and subsequently to damage to other cell components, mainly those of the microtubules and chromosome cohesive elements, leading to improper chromosome and chromatid separation during the first and the second meiotic division and, consequently, oocyte aneuploidy [43].

## 3. Update on Existing Clinical Strategies Evaluated in Women

From a practical viewpoint, the prevention and treatment of OA has to be directed to a variety of “unhealthy” conditions contributing to this pathology, including lifestyle, diet, environmental factors, and associated pathologies, along with their specific treatments. It is also important to make clear whether the intended strategy is aimed at fertility preservation or limited to the alleviation of the adverse effects of OA on the patient’s general health and wellbeing. The current clinical strategies are explained below, including antioxidant and mitochondrial therapies (Section 3.1), the use of GnRH agonists and antagonists (Section 3.2), hormones and growth factors (Section 3.3), and some more invasive treatments (Section 3.4).

### 3.1. Antioxidant and Mitochondrial Therapies

In general, antioxidant and mitochondrial therapies (Figure 2), aimed at slowing down OA, have proven to be both the most effective and best-tolerated therapies [44]. They have been shown to reduce oxidative stress caused by lifestyle factors such as smoking, unhealthy diet, drug abuse, or high alcohol consumption [45]. Among the variety of antioxidants available [11,42,43,44,46], a special position is reserved to melatonin. In fact, a synthesis of the recent data from both animal experimental and human clinical studies (reviewed in Tesarik and Mendoza Tesarik [47]) indicates that not only is melatonin a powerful antioxidant, acting both as a hormone and as a direct ROS scavenger; it also performs additional actions that benefit ovarian health, namely, anti-inflammatory and immunomodulatory activities (Figure 3). In fact, disequilibrium between individual elements of the immune system, mainly lymphocytes and macrophages, in addition to being involved in other gynecological pathologies, accelerate OA through a chronic, low-grade inflammatory state. At doses of 4–6 mg per day, melatonin is advisable both for delaying the onset of OA and for the treatment of women who already suffer from it [46,47].

### 3.2. GnRH Agonists and Antagonists

Other non-invasive, recently validated treatment approaches include pharmacological inhibition of follicle recruitment, hormonal and growth factor modulation, and the use of advanced mitochondrial-targeted antioxidants. Briefly, GnRH agonists and antagonists were used to inhibit follicular recruitment during anticancer chemotherapy, aiming at the protection of the ovarian follicular pool against the adverse effects of anticancer drugs. After early encouraging data obtained both in male patients [48] and in an experimental system using female monkeys [49], this strategy was applied to female cancer patients. However, several systematic reviews and meta-analyses have suggested that this approach may be controversial, and the outcomes may differ among the different cancer types and anticancer drugs used [50,51,52,53]. Further research into this subject is warranted, focusing on the individual clinical context, dosage optimization, and timing.

### 3.3. Hormones and Growth Factors

Hormonal and growth factor modulation has also been reported to give encouraging results in some patients facing OA. In fact, dehydroepiandrosterone (DHEA) supplementation led to improved serum ovarian reserve markers, such as inhibin B and antimullerian hormone (AMH), antral follicle count, as evaluated by a vaginal ultrasound scan, and in vitro fertilization outcomes [54]. Other recent (2023–2025) small randomized clinical trials have also highlighted the possibilities of using insulin-like growth factor 1 (IGF-1) and vascular endothelial growth factor (VEGF) to improve antral follicle counts (reviewed in Hirano et al. [13]), and further extensive clinical validation has been suggested to develop combination-based therapies (DHEA + IGF-1 + VEGF) by creating personalized formulas adapted to each individual biomarker profile, making further use of modern precision medicine. As early as 2005, growth hormone (GH), supposedly acting—at least partially—through IGF-1, was shown to improve IVF outcomes in women aged > 40 years when administered during ovarian stimulation [55], and subsequent studies basically confirmed these findings and suggested the underlying mechanisms of GH action [56,57]. New formulations of GH delivery systems, based on novel biomaterials (see Section 4.1), have been shown to improve the bioavailability of GH and may lead to innovative therapeutic strategies for preventing and treating ovarian dysfunction [58]. Nonetheless, these treatments still have limitations. In particular, while the short-term beneficial effects on oocyte quality were demonstrated, further studies are needed to confirm their long-term effects in humans, including the absence of negative consequences for offspring health.

### 3.4. More Invasive Treatments

In addition to the above non-invasive treatments, other more invasive treatments are also currently in use. The intra-ovarian injection of platelet-rich plasma (PRP), prepared from the patient’s own blood, has shown promising results in promoting the activation of dormant primordial follicles, which is believed to be due to the action of platelet-derived growth factors [59]. Fertility preservation, either through oocyte recovery and freezing or ovarian tissue cryopreservation in view of subsequent autotransplantation, is another available option [60]. In addition to being used for restoring fertility, ovarian tissue cryopreservation and autotransplantation may also be considered for endocrine function restoration [61]. Mitochondrial donation, carried out via pronuclear transfer, has recently been shown to be effective (8 live births out of 22 attempts) in women with pathogenic mtDNA variants [62]. Other advanced strategies, tested in experimental animal models, are described below (Section 4).

## 4. Outlook

Recently, a number of new therapeutic strategies for OA, though still not ready for clinical application, have emerged. There is a lot of new experimental data, obtained by experimental studies mainly performed in rodents, that could lead to the development of clinically available solutions for slowing down natural or premature OA in the near future. New developments in OA treatment strategies are based on our knowledge of the molecular players governing this process and our understanding of how they can be efficiently targeted by specific therapeutic approaches. Particular attention has been paid to biomedicine strategies, such as biomaterials, nanoparticles, extracellular vesicles, and “intelligent” advanced drug carriers capable of sensing natural physiological processes so as to achieve controlled drug release for patient-tailored ovarian microenvironment reprogramming, tissue repair, and immune and metabolic regulation (reviewed in Liang et al. [63]). This section deals with the potential use of new biomaterials (Section 4.1), metal-based biomedicines (Section 4.2), and natural biomedicines (Section 4.3). Perspectives and precautions in view of the future clinical strategies are resumed in Section 4.4.

### 4.1. Biomaterials

Research into new biomaterials in regenerative medicine, including those recreating the three-dimensional environment for in vitro cultured cells, has been aimed at creating a microenvironment favoring cell interactions, promoting good passive and active targeting, and allowing for high drug loading capacity and controlled drug release, while still displaying good stability and biodegradability. Two types of biomaterials have been investigated in animal models of OA—natural ones and synthetic ones, each showing its own advantages and disadvantages [64]. Briefly, natural biomaterials, derived from biological sources like humans, plants, animals, and microorganisms, are valued for being biocompatible and biodegradable and can be categorized based on their composition, i.e., as protein-based and polysaccharide-based.

In the context of OA prevention and therapy, recent research highlights several types of natural biomaterials, including stem cell-derived extracellular vesicles, human amniotic cell-derived, follicular fluid-derived and menstrual blood stromal cell-derived exosomes, the decellularized extracellular matrix, collagen, hyaluronic acid, fibrin, and alginate. All of these have been evaluated and have shown promising results in animal models. Likewise, synthetic (polyethyleneglycol) supramolecular hydrogels were tested in animal models of OA, with promising results (reviewed in Wu et al. [64]).

### 4.2. Metal-Based Biomedicines

Nanoparticles (NPs) based on cerium dioxide and silver have attracted widespread attention due to their antioxidant and anti-inflammatory action, likewise silver and magnesium oxide NPs, which, in addition, were shown to alleviate the hormone imbalances occurring during OA. In turn, zinc oxide NPs have not only exhibited antioxidant and anti-inflammatory effects but have also been shown to be particularly useful in the context of OA associated with diabetes, enhancing insulin sensitivity and regulating glucose metabolism (reviewed in Wu et al. [64]).

### 4.3. Natural Biomedicines

Knowledge derived from ancestral natural medicine formulas has recently been re-evaluated in animal models of OA. For example, celastrol, a compound extracted from the Chinese plant *Triprerygium wilfordii*, has been shown to promote ovarian follicle development in mice and pigs by regulating granulosa cell proliferation and apoptosis [65]. A 2-month treatment of mice with low-molecular-weight chitosan, a natural fibrous polysaccharide derived from chitin, led to delayed OA, and this effect was apparently mediated by enhanced macrophage phagocytosis and improved ovarian tissue homeostasis [66]. Another study has demonstrated that collagen extracted from the sturgeon swim bladder can counteract OA in mice exposed to cyclophosphamide through a variety of mechanisms, including antioxidant effects, the activation of the PI3K/Akt and Bcl-2/Bax pathways, and the inhibition of the mitogen-activated protein kinase pathway, thereby reducing apoptosis [67].

### 4.4. Perspectives and Precautions

The analysis of the mechanisms causing ovarian cellular senescence, including the accumulation of advanced glycation end products, oxidative stress, mitochondrial dysfunction, DNA damage, telomere shortening, and exposure to chemotherapy in six distinct cell types (oocytes, granulosa cells, theca cells, immune cells, ovarian surface epithelium, and endothelial cells), has suggested the potential of ovarian senotherapeutics in the treatment of OA [68]. They include the use of personalized (i.e., adapted to the current condition of each individual) mixtures of specific agents targeting the key factors involved in the senescence of different ovary cell types (see above), including desatinib, quercetin, rapamycin, metformin, resveratrol, melatonin, and coenzyme Q10 [68]. New vehicles of GH administration, facilitating its bioavailability, are also under investigation (see Section 3). However, it has to be stressed that all these therapies are currently in the pre-clinical stage, and their application in humans will require more studies concerning their short- and long-term safety.

The employment of new biomaterials and biomedicines, though showing promising results in animal experiments, awaits further validation before clinical application with regard to the efficacy and safety of these materials in humans. Above all, taking into account the basic principal of “first do not harm,” a fundamental ethical principle in medicine, the eventual clinical use of these therapies should follow a thorough review of their potential effects in terms of both short-term and long-term offspring health.

Finally, the current trend towards repeating in vitro fertilization/ovarian stimulation cycles has aroused concern about possible long-term adverse effects on the ovarian reserve (i.e., the stock of oocytes in the ovaries). This issue was addressed by a recent review [69]. The data presented suggest that repeated ovarian stimulation can induce changes in the immune response and increase oxidative stress in the ovarian microenvironment, leading to an accelerated loss of ovarian reserve. This risk further explains the current interest in the use of melatonin (see Section 3.1) before, during, and after ovarian stimulation (see Section 3.1).

## 5. Conclusions

OA is a gradual process whereby several elements of the HPO axis progressively deteriorate, leading to disturbances in the activity of different types of ovarian cells, dysregulation of the menstrual cycle, decreased oocyte quantity and quality, and impaired ovarian endocrine functions. While the oocyte issues compromise fertility, the endocrine problems have consequences that go further than reproduction, increasing propensity to certain chronic diseases and negatively affecting the general health status and wellbeing of the individual. Physiologically, some processes involved in OA start at ages 25–30 and accelerate significantly between the ages of 35 and 40, ultimately leading to complete menopause after 50 years of age. However, OA may start prematurely in some women and can easily go undetected until the possibilities of preventive and therapeutic interventions are diminished. Therefore, subtle signs of premature OA should be looked for, especially in women with a genetic background that predisposes them to this pathology. If the ovaries can still be expected to contain sufficient healthy cells, fertility and endocrine function preservation through ovarian tissue cryopreservation for later autotransplantation may be envisaged.

Preventive and therapeutic actions against OA include those targeting the HPO axis and those acting directly on the ovary. Suppression of ovarian follicular development, either acting at the receptor level in hypothalamic GnRH-secreting neurons or directly by using GnRH agonists and antagonists, was largely successful in animal experiments, and some of these treatments are currently being introduced into clinical practice. Direct action at the ovary level has an even longer tradition; it is based on the oral intake of melatonin and other antioxidants or makes use of hormones and GFs (GH, DHEA, IGF-1, VEGF). The most recent findings point to the development of precision medicine approaches based on personalized combinations of different agents, adapted to each individual patient’s condition.

## Figures and Tables

**Figure 1 ijms-26-11973-f001:**
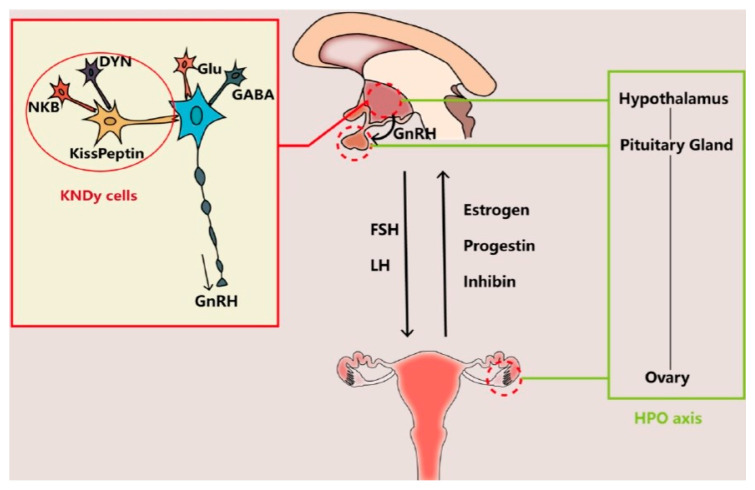
The interaction between the brain and the ovary through the HPO axis. The main players regulating the release of gonadotropin-releasing hormone (GnRH) from the hypothalamus include glutamate (Glu), gamma-aminobutyric acid (GABA), and transmitters secreted by KNDy cells, co-expressing kisspeptin, neurokinin B (NKB), and dynorphin (DYN). The brain interacts with the ovary through hormones secreted from the pituitary gland (FSH and LH) and the ovary (estrogen, progestin, and inhibin). Reproduced from Wang et al. [11]. © The authors, in terms of the Creative Commons Licence Agreement.

**Figure 2 ijms-26-11973-f002:**
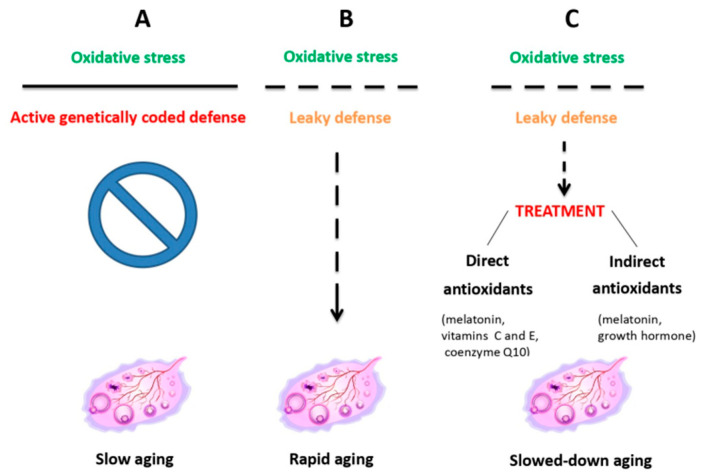
Ovarian aging due to the oxidative stress resulting from mitochondrial dysfunction is counteracted by inherent antioxidant defenses, which, when leaky, can be helped by external antioxidant administration. Reproduced from Tesarik et al. [44]. © The authors, in terms of the Creative Commons Licence Agreement.

**Figure 3 ijms-26-11973-f003:**
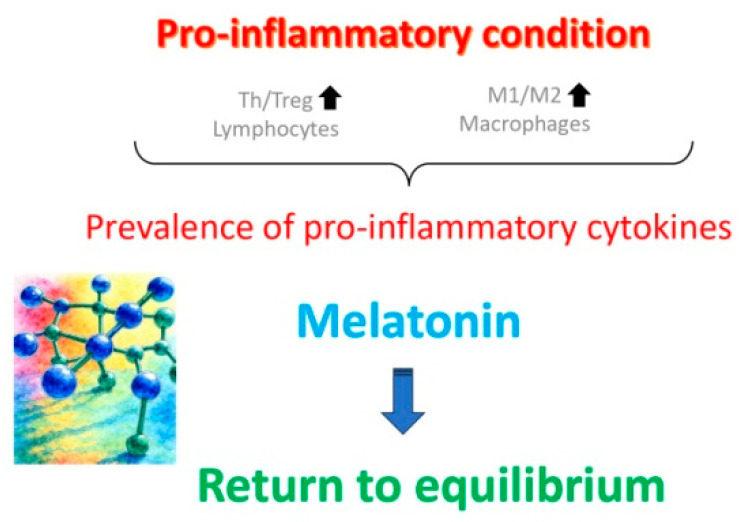
Ovarian aging is marked, in particular, by increased prevalence of T helper (Th) lymphocytes over T regulatory (Treg) lymphocytes, along with an increased ratio of M1/M2 macrophages in the uterine cavity, with the consequent prevalence of pro-inflammatory cytokines maintaining chronic, low-grade inflammatory state. Melatonin treatment reverts these anomalies and helps slow down the aging process. Reproduced from Tesarik and Mendoza Tesarik [47]. © The authors, in terms of the Creative Commons Licence Agreement.

**Table 1 ijms-26-11973-t001:** Summary of existing strategies to counteract ovarian aging.

Targeting HPO Axis	Targeting Ovarian Cells		
Glutamate receptors	Antioxidants		Hormones and GFs
GABA receptors			
GnRH agonists	Direct	Indirect	GH
GnRH antagonists			DHEA
	Melatonin	Melatonin	IGF-1
	CoQ10		VEGF
	Vitamin C		PRP (PDGF)
	Vitamin E		
	Folic acid		
	Others		

Abbreviations: HPO: hypothalamic–pituitary–ovarian axis; GFs: growth factors; GABA: gamma-aminobutyric acid; GnRH: gonadotropin-releasing hormone; GH: growth hormone; DHEA: dehydroepiandrosterone; IGF-1: insulin-like growth factor 1; CoQ10: coenzyme Q10; VEGF: vascular epidermal growth factor; PRP: platelet-rich plasma; PDGF: platelet-derived growth factor.

## Data Availability

No new data were created or analyzed in this study. Data sharing is not applicable to this article.

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
