# Peer review of "Clinical Strategies for Counteracting Human Ovarian Aging: Molecular Background, Update, and Outlook"

_ijms, 2025, doi:10.3390/ijms262411973_

Round 1
Reviewer 1 Report
Comments and Suggestions for Authors
This review systematically delineates the complex mechanisms of ovarian aging and the corresponding clinical countermeasures, and outlines future therapeutic directions. Its central thesis is that effective intervention requires a comprehensive, individualized precision-medicine strategy that concurrently targets the central nervous system (the hypothalamic–pituitary–ovarian axis, HPO axis) and the cellular senescence processes within the ovary itself. The following comments are offered to improve the quality of the manuscript:
- The article structure is unclear and transitions between sections are abrupt; consider adding bridging sentences at the beginning of each section.
- The abstract is overly general and does not explicitly state the unique contributions of this review compared with existing ones.
- In the antioxidant therapy section, melatonin is highlighted as “the most effective,” yet direct comparative data with other antioxidants (e.g., CoQ10, vitamins) are lacking.
- The “personalized cocktail therapy” mentions agents such as dasatinib and rapamycin, for which clinical studies in ovarian aging are scarce; the current research stage and potential risks should be clarified.
- Figures 1–3 are reproduced from other sources; the review contains no original schematic. A summary table or mechanistic diagram of existing strategies is recommended.
- Some references are formatted inconsistently (e.g., mixed use of abbreviated and full journal names); please standardize according to the journal’s requirements.
Author Response
Dear Reviewer
Thank you very much for your comments which will certainly increase the manuscript quality. I have included the corresponding modifications in the manuscript (highlighted in yellow). This is a list of the changes made.
- The article structure is unclear and transitions between sections are abrupt; consider adding bridging sentences at the beginning of each section.
Response: Bridging sentences have been added at the end of each section to introduce the following subsections.
- The abstract is overly general and does not explicitly state the unique contributions of this review compared with existing ones.
Response: Two sentences pointing out the unique character of this review have been added at the end of Abstract.
- In the antioxidant therapy section, melatonin is highlighted as “the most effective,” yet direct comparative data with other antioxidants (e.g., CoQ10, vitamins) are lacking.
Response: In fact, there is no direct data comparing the antioxidant effectiveness of melatonin with that of other antioxidants. Melatonin is highlighted because its antioxidant action is both direct (ROS scavenger) and indirect (receptor-mediated) and, especially, because it also produces important immunomodulatory and anti-inflammatory effects, in addition to acting as antioxidant. This has been made clear in the revised manuscript, and the expression “the most effective antioxidant” has been removed.
- The “personalized cocktail therapy” mentions agents such as dasatinib and rapamycin, for which clinical studies in ovarian aging are scarce; the current research stage and potential risks should be clarified.
Response: At the end of the first paragraph of subsection 4.4. a sentence has been added stating that it has to be stressed that all these therapies are currently at the pre-clinical stage, and their application in humans will need more studies as to their short- and long-term safety.
- Figures 1–3 are reproduced from other sources; the review contains no original schematic. A summary table or mechanistic diagram of existing strategies is recommended.
Response: A summary table of the currently existing strategies for counteracting ovarian aging has been added.
- Some references are formatted inconsistently (e.g., mixed use of abbreviated and full journal names); please standardize according to the journal’s requirements.
Response: The reference formatting has been stardardized according to the journal’s requirements.
Best regards
Jan Tesarik
Reviewer 2 Report
Comments and Suggestions for Authors
Comments:
The authors present a focused and comprehensive review of the factors contributing to ovarian aging and how various influences affect germ cells throughout this process. The manuscript also provides up-to-date insights into potential preventive strategies against ovarian aging. It is well written, covers most key aspects of ovarian aging, and includes relevant clinical perspectives. Both authors have done an excellent job, and the article is suitable for publication in this journal.
I have a few suggestions that could further strengthen the scientific depth of the paper:
- Including a discussion of the limitations associated with commonly used approaches to improve germ-cell quality—such as growth factor–based interventions—would be valuable, particularly because these methods can influence ovarian reserve, which is closely linked to ovarian aging.
- It would also strengthen the review to address how ART-related superstimulation/superovulation protocols affect germ-cell quality and how these interventions may impact ovarian reserve in ways connected to ovarian aging.
Author Response
Dear Reviewer
Thank you very much for your comments which will certainly increase the manuscript quality. I have included the corresponding modifications in the manuscript (highlighted in yellow). This is a list of the changes made.
- Including a discussion of the limitations associated with commonly used approaches to improve germ-cell quality—such as growth factor–based interventions—would be valuable, particularly because these methods can influence ovarian reserve, which is closely linked to ovarian aging.
Response: A brief discussion of the limitations associated with the use of growth factor-based interventions, particularly with regard to long-term effects on the ovarian reserve and the potential health risks for offspring, has been added at the end of subsection 3.3.
- It would also strengthen the review to address how ART-related superstimulation/superovulation protocols affect germ-cell quality and how these interventions may impact ovarian reserve in ways connected to ovarian aging.
Response: A couple of sentences (at the end of subsection 4.4) and a new reference (72) have been added to address this issue.
Best regards
Jan Tesarik